# Impact of hydronium ions on the Pd-catalyzed furfural hydrogenation

Iris K. M. Yu [1,2], Fuli Deng[1], Xi Chen[1], Guanhua Cheng[1,3], Yue Liu [1,4], Wei Zhang[1,4] ✉ & Johannes A. Lercher [1,5] ✉

In aqueous mediums, the chemical environment for catalytic reactions is not only comprised of water molecules but also of corresponding ionized species, i.e., hydronium ions, which can impact the mechanism and kinetics of a reaction. Here we show that in aqueous-phase hydrogenation of furfural on Pd/C, increasing the hydronium ion activities by five orders of magnitude (from pH 7 to pH 1.6) leads to an increase of less than one order of magnitude in the reaction rate. Instead of a proton-coupled electron transfer pathway, our results show that a Langmuir-Hinshelwood mechanism describes the rate-limiting hydrogen addition step, where hydrogen atom adsorbed on Pd is transferred to the carbonyl C atom of the reactant. As such, the strength of hydrogen binding on Pd, which decreases with increasing hydronium ion concentration (i.e., 2 kJ $mol_{H2}^{-1}$ per unit pH), is a decisive factor in hydrogenation kinetics (rate constant +270%). In comparison, furfural adsorption on Pd is pH-independent, maintaining a tilted geometry that favors hydrogen attack at the carbonyl group over the furan ring.

Hydrogen binding energy (HBE) on platinum group metals plays a critical role in catalytic hydrogenation of, for example, benzaldehyde or phenol and are of broad interest for developing a detailed mechanistic understanding of this important class of reactions[1,2]. In aqueous environment, electrochemical analyses show that HBE tends to decrease with solution pH[3,4], plausibly due to changing electronic properties of metal nanoparticles, the organization and, hence, standard free energy of water adsorption, as well as cation concentration at the electrolyte-metal interface[5–7]. Therefore, the dependence of HBE on hydronium ion activities is highly relevant to the kinetics of aqueous-phase hydrogenation. For instance, the pH-induced decrease in HBE reduces the energy difference between the adsorbed state and transition state, leading to a lower intrinsic activation energy for hydrogen addition to phenol on Pt and, consequently, increasing the hydrogenation rate by over an order of magnitude with the change

from pH 8 to pH 1[2]. While the impact of pH on HBE and the consequence on hydrogenation rate is explored mostly using monofunctional compounds, e.g., phenol and benzaldehyde[1,2], less is known about how it impacts the kinetics and thermodynamics of the reaction of more reactive, highly functionalized molecules. The latter allows several parallel hydrogenation pathways that could result in a mix of products with low selectivity.

Furfural is chosen as a model compound in this study to gain fundamental insights for such more complex reactants. It is produced from lignocellulosic biomass as an abundant renewable resource, drawing significant attention as a versatile platform molecule to serve wide applications in a carbon-neutral future[8–13]. Furfuryl alcohol (FAL) is one of the important hydrogenation products of furfural and can be used as the monomer for producing FAL resin[14]. In addition, FAL can be subjected to upgrading processes to produce methyl furan as a fuel

[1]Department of Chemistry and Catalysis Research Center, Technische Universität München, Lichtenbergstrasse 4, 85748 Garching, Germany. [2]Research Institute for Future Food and Department of Applied Biology and Chemical Technology, The Hong Kong Polytechnic University, Hung Hom, Kowloon, Hong Kong, China. [3]Key Laboratory for Liquid–Solid Structural Evolution and Processing of Materials (Ministry of Education), School of Materials Science and Engineering, Shandong University, Jingshi Road 17923, Jinan 250061, China. [4]Shanghai Key Laboratory of Green Chemistry and Chemical Processes, School of Chemistry and Molecular Engineering, East China Normal University, Shanghai 200062, China. [5]Institute for Integrated Catalysis, Pacific Northwest National Laboratory, P.O. Box 999 Richland, WA 99352, USA. ✉e-mail: weizhang@chem.ecnu.edu.cn; johannes.lercher@tum.de

additive and levulinic acid as a precursor to green solvents, fuels, polymers, etc[15,16].

From a scientific viewpoint, the high reactivity of furanic molecules provides an excellent opportunity to understand determinants of catalytic activity and product selectivity. Hydrogen attack on the furfural carbonyl C=O group and/or on the ring C=C double bond have been frequently reported, producing FAL and tetrahydrofurfural (THFF), respectively, depending on the catalysts and solvents used[17–19]. We presume a fundamental driving force for hydrogenation pathway that is related to the co-adsorption of oxygenates and $H_2$, based on the knowledge of adsorption mode-directed selectivity[10,20]. Previous density functional theory (DFT) calculations suggest a strong coverage- and metal-dependence of furfural adsorption[10,20]. However, the literature is lacking the understanding of impact of co-adsorbates on one another as well as the corresponding implications in aqueous-phase catalysis. In this study, we scrutinize $H_2$ adsorption in parallel with furfural binding and the associated impacts on hydrogenation kinetics and thermodynamics.

Water is reported to be a superior medium for the hydrogenation of furfural and more facile FAL formation was observed in aqueous phase than in cyclohexane. DFT calculations suggest a water-mediated pathway with significantly lower energy barrier, where water stabilizes the transition states and participates in a hydrogenation step following a proton-coupled electron transfer (PCET) pathway[21]. This leads to the more general question about the impact of the aqueous medium on hydrogenation kinetics. Previous electrochemical analysis indicates that electrolytes at low pH favor water adsorption and electrolysis on Pt(100)[5]. In the absence of an external electric potential, however, more subtle pH effects are expected for hydrogenation, as the open circuit potential will vary in a more muted way, which in turn will lead to subtler influences on the thermodynamic states of all reacting partners.

In this work, the Pd-catalyzed aqueous-phase hydrogenation of furfural is investigated to probe the impact of the hydronium ion concentration on the state of hydrogen and organic reactant binding and consequentially on the catalytic activity and selectivity. Our results suggest that a tilted position of adsorbed furfural is selectively hydrogenated at the aldehyde group rather than the unsaturated furan ring. The rate-determining step in the hydrogenation proceeds via the addition of Pd-adsorbed hydrogen to carbonyl C of furfural rather than via PCET as recently suggested[21,22]. The hydrogen binding strength is a critical determinant of the intrinsic energy barrier for hydrogenation in the concerned kinetic regime ($0^{th}$ reaction order in furfural and $1^{st}$ order in $H_2$). The way the solvent impacts kinetics and thermodynamics via altering the binding of dual adsorbates inspires a new perspective to design bimolecular reaction systems.

## Results and discussion

### Dependence of the rate of furfural hydrogenation on hydronium ion activity

The pH effect on the hydrogenation of furfural to furfuryl alcohol (FAL) was examined in phosphate buffer solutions on the Pd/C catalyst at room temperature. The conversion increases linearly ($R^2 > 0.99$) with reaction time, even up to high conversions of 70 mol% (Fig. 1a). Figure 1b shows that the hydrogenation rate increases with the decreasing pH in general (selectivity and carbon balance in Table S1). That is, when the buffer changes from pH 7 to pH 1.6, the turnover frequency (TOF) of FAL formation ($mol_{FAL}\ mol_{Pd}^{-1}\ h^{-1}$) increased from 287 to 2422 $h^{-1}$ at 10 bar $H_2$. Similar acid-promoted hydrogenation of phenol and benzaldehyde have been reported in previous studies despite the use of different catalysts and electrocatalytic methods[2,23]. All findings together with the present results are strong evidence that the concentration of hydronium ions generally promotes hydrogenation, regardless of the specific mechanism. Such promotion has been frequently associated with a (gradual) change in the mechanism of hydrogen addition.

Hydrogenation in the absence of external electric potential tends to follow the Langmuir Hinshelwood (LH) mechanism, where molecular $H_2$ dissociatively adsorbs on a metal surface, forming adsorbed H atoms (H*) that react with the co-adsorbed organic reactant[24,25]. Proton-coupled electron transfer (PCET) as an alternative pathway involves the addition of protons (from solvent) and electrons (from metal catalyst) to reactants. Although PCET is commonly considered for electrochemical reactions, it has also been proposed for reactions occurring without an external electric potential[21,22]. We explore its possibility in the present work, in view of the increasing discussion of water[21] and other protic solvents[22,26] participating in hydrogen addition steps. Proton-coupled electron transfer has been hypothesized to be more prominent at low pH because of the higher $H_3O^+$ activity.

To test the hypothesis, we use the kinetic isotope effects (KIEs) on furfural hydrogenation at room temperature. The four sets of conditions include using $D_2O$ as the solvent coupled with $H_2$ gas ($H_2/D_2O$), $D_2/D_2O$, $D_2/H_2O$, and $H_2/H_2O$. Changing the solvent from $H_2O$ to $D_2O$ has a miniscule impact on the TOFs for furfural hydrogenation at both pH 1.6 and 5.8 (Fig. 2a). In comparison, the rates decrease approximately by 50% as $D_2$ is used instead of $H_2$, suggesting that $H_2/D_2$ from gas phase plays a more important role than $H^+/D^+$ from water in the rate-determining step (rds).

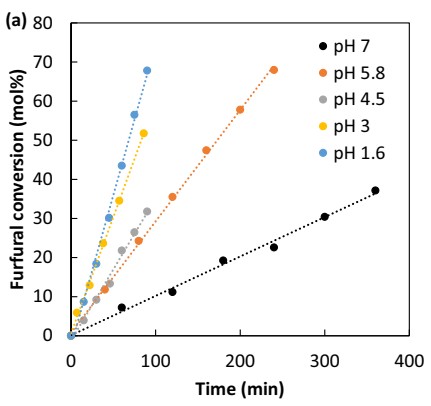
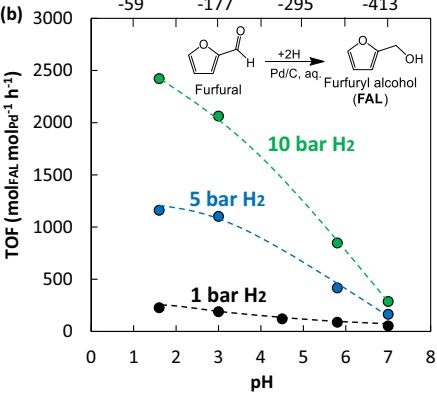
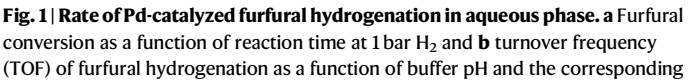

**Fig. 1 | Rate of Pd-catalyzed furfural hydrogenation in aqueous phase. a** Furfural conversion as a function of reaction time at 1 bar $H_2$ and **b** turnover frequency (TOF) of furfural hydrogenation as a function of buffer pH and the corresponding open circuit potentials at different pressures of $H_2$. The reaction was performed with 30 mM furfural in 0.1 M phosphate buffer solution with 10 mg Pd/C at 1–10 bar $H_2$ and room temperature. Source data are provided as a Source Data file.

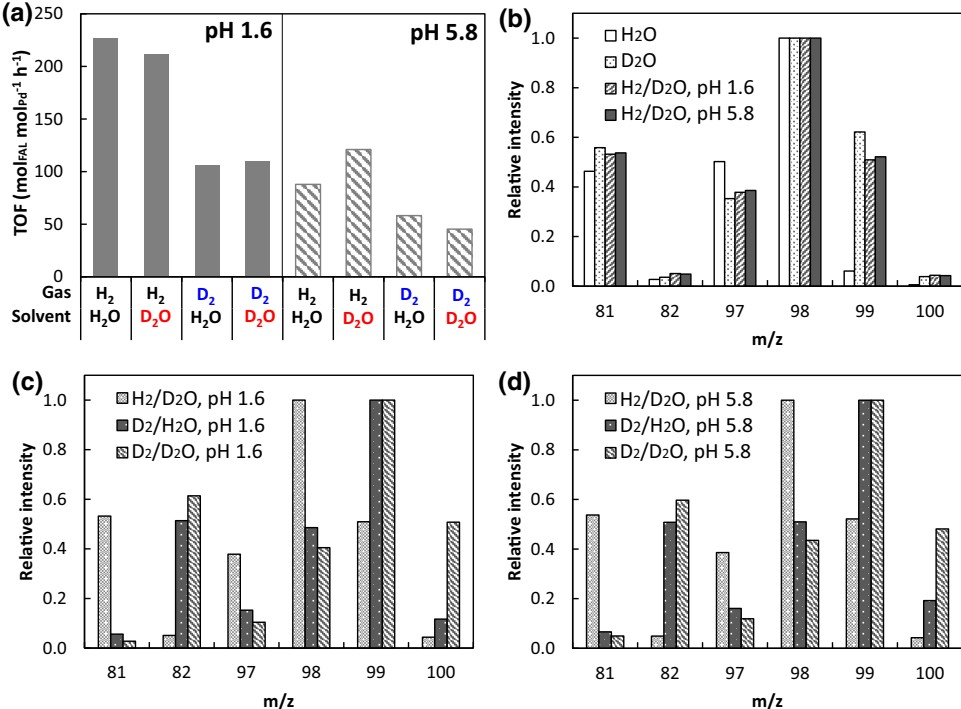

**Fig. 2 | Kinetic isotope effects on Pd-catalyzed hydrogenation in aqueous phase. a** Turnover frequency (TOF) of furfural hydrogenation in $D_2O/H_2O$ and $D_2$/ $H_2$ and **b–d** the mass spectra of the produced FAL. Mass spectra of **b** standard FAL dissolved in $H_2O$ and $D_2O$ in comparison with those of FAL produced form furfural in $H_2$; and mass spectra of FAL produced from furfural at **c** pH 1.6 and **d** pH 5.8. Furfural hydrogenation was performed with 30 mM furfural in 0.1 M phosphate buffer solution ($D_2O$ or $H_2O$) with 10 mg Pd/C at 1 bar $D_2$ or $H_2$ and room temperature. Source data are provided as a Source Data file.

To trace deuterium incorporated in the product, the produced FALs were also analyzed by gas chromatography-mass spectrometry (GC/MS). The MS patterns of FAL produced under $H_2/D_2O$ at both pH 1.6 and 5.8 are similar to that of a standard solution of commercial FAL in $D_2O$ (Fig. 2b). This indicates rapid H/D exchange of FAL after its formation in $D_2O$, rather than any D transfer from solvent to furfural during the reaction. Changing the gas phase from $H_2$ to $D_2$, significant increases in the weight of molecular ion and fragment ions were observed, indicating the incorporation of D to the product FAL (Fig. 2c, d). These results suggest molecular $H_2$ (and $D_2$) as the major reducing agent for furfural hydrogenation irrespective of pH. Therefore, we conclude that in the rate-determining step, hydrogen (rather than a proton) is added and that the reaction follows a LH mechanism.

It is noteworthy that both acidic (pH 1.6) and close to neutral (pH 5.8) conditions give the same isotope tracing results (Fig. 2) and the same reaction order of one in $H_2$ (in $H_2O$; Fig. S1a), suggesting that the mechanism remains unchanged across the studied pH range. In particular, the 1st order in $H_2$ indicates that the second H addition, very likely to the C atom of the furfural carbonyl group[21], is the rds, according to the derived kinetic equations (details in Supplementary Note S4). Note that we cannot infer from the KIE study that whether the first H addition step, which is quasi-equilibrated, follows the LH pathway or PCET. The hydrogenation rate ($r$) therefore depends on the parameters below:

$$r = k_2 \theta_H \theta_{FH} \tag{1}$$

where $k_2$ is the rate constant in the rds, $\theta_{FH}$ is the coverage of the intermediate−furfural with one H added, and $\theta_H$ is the coverage of H* on Pd surface. We then investigate, which of these parameters are pH-sensitive and how they contribute to the pH-dependence of furfural hydrogenation. The apparent activation energy at pH 1.6 and pH 5.8 were measured to be similar ($9.9 \pm 1.6$ kJ mol$^{-1}$ and $13.8 \pm 1.2$ kJ mol$^{-1}$,

respectively; Fig. S2). This energy barrier mainly describes the energy level difference between the transition state and the state of adsorbed furfural (initial state), considering the 0th order in furfural and 1st order in $H_2$ (Fig. S1). Therefore, we hypothesize that in such a bimolecular surface reaction, $H_2$ rather than furfural plays a more dominant role in determining the hydrogenation kinetics.

**Furfural adsorption**

The adsorption isotherms show that changing the pH does not affect the adsorption of furfural on Pd/C (Fig. 3a). By fitting with the Langmuir isotherm equation, the adsorption equilibrium constant ($K_F^o$) is determined as $2297 \pm 282$ at pH 1.6–7. This corresponds to a standard Gibbs free energy ($\Delta G^o$) of adsorption of about −19 kJ mol$^{-1}$ at room temperature. The interesting consequence is that furfural−metal interactions are not strongly impacted by the activity of surrounding $H_3O^+$ or changes in the double layer properties induced by varying hydronium ion concentrations[5]. In Fig. 3b, the enthalpy of furfural adsorption ($\Delta H_{F\,ads,aq}^0$) is measured to be $-28 \pm 5$ kJ mol$^{-1}$ at pH 5.8 using liquid calorimetry, which is close to the value at pH 3 ($-30.0 \pm 4.6$ kJ mol$^{-1}$) and in pure water ($-31 \pm 3$ kJ mol$^{-1}$). It is conceivable that in the studied system, the furfural-Pd bond in aqueous phase remains unaffected by electrochemical changes associated with the increasing hydronium ion activity. It also suggests that the binding is not affected by the changing open circuit potential (as induced by the variations in pH). The furfural concentration of 30 mM in most hydrogenation experiments in this work lies in the saturation region of the adsorption isotherms (Fig. 3a), which is consistent with the measured zeroth order (Fig. S1b). These findings together suggest furfural saturation on the catalyst surface so that further increase in furfural concentration does not impact the reaction rate.

More insights are needed on the furfural/Pd interface under experimental condition. Without the influence of water (i.e., in ultra-high vacuum (UHV)), the furfural adsorption enthalpy ($\Delta H_{F\,ads,g}^0$) was

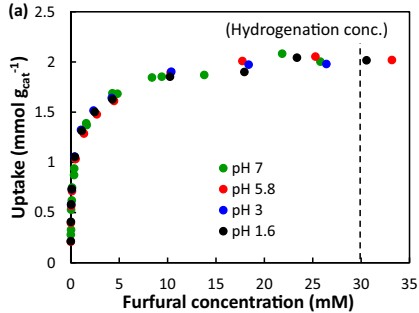
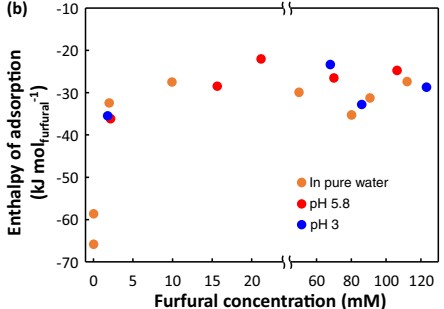

**Fig. 3 | Furfural adsorption on Pd/C in aqueous phase. a** Isotherms of furfural adsorption on Pd/C in 0.1 M phosphate buffer solution at room temperature (dashed line indicates the furfural concentration used in the hydrogenation reaction). **b** Adsorption enthalpy of furfural adsorption on Pd/C as a function of furfural concentration in aqueous phase. The measurement was performed using liquid calorimetry at room temperature. Source data are provided as a Source Data file.

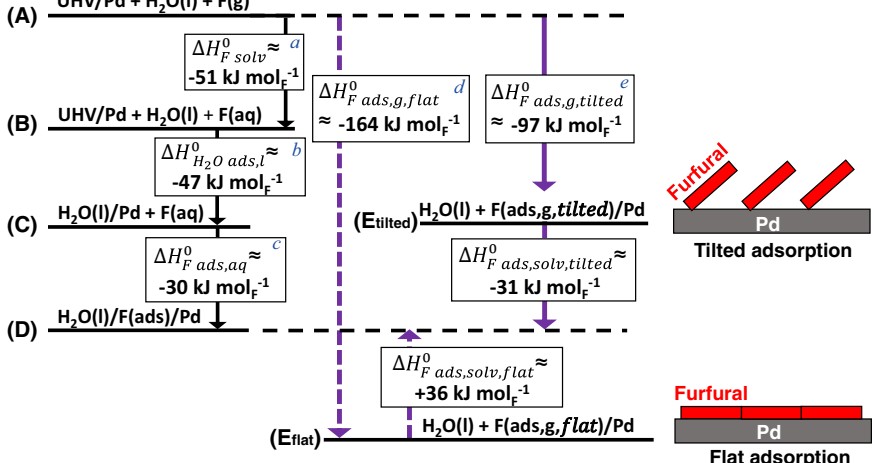

**Fig. 4 | Thermodynamic cycle for the gas- and aqueous-phase adsorption of furfural (F) on Pd.** A replacement ratio of 6.5 water molecules by one furfural molecule on Pd is used. Ultrahigh vacuum above the clean Pd surface is denoted as UHV/Pd. For simplicity, the difference between bond energy ($\Delta U$) and enthalpy ($\Delta H$) in gas-forming steps is considered negligible ($\Delta U = \Delta H + RT$, and $RT = 2.5$ kJ mol$^{-1}$ at 298 K is omitted). **a** Solvation enthalpy calculated using the van't Hoff equation and Henry's law constant[27,28]. **b** Enthalpy of adsorption of water on Pd (see Supplementary Note S1). **c** Enthalpy of saturated adsorption in aqueous phase measured by liquid colorimetry. **d** Computed adsorption enthalpy at 0.25 ML and **e** at 0.5–1 ML in UHV from ref. [20].

computed to be about −164 kJ mol$^{-1}$ (−1.7 eV), if furfural is adsorbed in parallel to the Pd surface at a low density of 0.25 monolayer (ML)[20]. The enthalpy changes to about −97 kJ mol$^{-1}$ as the surface becomes crowded at 0.5–1 ML and furfural binds to Pd via the −CHO group, resulting in a tilted conformation[20]. Both reported $\triangle H^0_{F\,ads,g}$ are more negative than our calorimetric measurement in aqueous phase ($\triangle H^0_{F\,ads,aq} \approx -30$ kJ mol$^{-1}$ for saturated adsorption).

The question arises about whether furfural takes the flat or tilted adsorption mode under the experimental hydrogenation conditions (i.e., 30 mM, saturation region). To answer, we construct the thermodynamic cycle for furfural solvation and adsorption in gas phase and aqueous phase (Fig. 4). It involves changes from state (A) to (B)—furfural solvation, and from state (B) to (C)—H$_2$O adsorption on Pd, of which the enthalpy calculation details are provided in Supplementary Note S1. The change from state (C) to (D) refers to the adsorption of furfural(aq) on Pd, of which the enthalpy ($\triangle H^0_{F\,ads,aq}$, ~ 30 kJ mol$^{-1}$) measured by liquid calorimetry under saturation region was applied here. The change from state (A) to ($E_{flat}$) or ($E_{tilted}$) is the adsorption of gas furfural on Pd in the flat or tilted mode, respectively. The DFT calculated enthalpy ($\triangle H^0_{F\,ads,g}$) of parallel and tilted adsorption was about −164 kJ mol$^{-1}$ and −97 kJ mol$^{-1}$, respectively[20]. To close the thermodynamic cycle, changing from state ($E_{flat}$) to (D) would be associated with an enthalpy $\triangle H^0_{F\,ads,solv,flat}$ of +36 kJ mol$^{-1}$. In comparison,

changing from state ($E_{tilted}$) to (D) would be an exothermic process with the enthalpy $\triangle H^0_{F\,ads,solv,tilted}$ of −31 kJ mol$^{-1}$. The step from either ($E_{flat}$) or ($E_{tilted}$) to (D) is essentially the solvation of adsorbed furfural by liquid water and, thus, is more reasonable to be exothermic. Therefore, adsorbed furfural molecule is more likely to be in the tilted geometry at saturated region, which favors H addition at the carbonyl group, in agreement with the formation of FAL as the primary product instead of THFF (ring hydrogenation product) in aqueous-phase Pd-catalyzed hydrogenation (Table S1).

The conclusion that furfural adsorbs (on Pd) at saturation in a tilted mode is established on the basis of coverage dependency. Taking a step further, one will expect that the adsorption will shift from the tilted mode to flat mode as the furfural concentration decreases. As we hypothesized, Fig. 3b shows that the adsorption enthalpy becomes more negative, when the concentration of furfural drops to 2 mM and below, e.g., −66 kJ mol$^{-1}$ at 0.03 mM furfural. The more exothermic binding implies the prevalence of parallel adsorption mode that facilitates stronger furfural-Pd interaction. In addition, the product selectivity of furfural hydrogenation shifts from carbonyl hydrogenation product (FAL) to ring hydrogenation product (THFF) when the furfural concentration decreases from 30 mM to 1.3 mM, i.e., leading to the increase in THFF-to-FAL ratio from 0.15/1 to 0.8/1 (Fig. S3 and Table S2). These findings substantiate the fact that furfural coverage

plays critical role in controlling the adsorption mode and, hence, potentially the hydrogenation selectivity.

## Hydrogen adsorption

The 1st order in $H_2$ suggests a rather low $\theta_H$ under the studied conditions (Fig. S1a). To evaluate the pH effect on $H_2$ adsorption ($H_2(g) + 2^* \rightarrow 2H^*$), we performed H/D exchange between $H_2$ gas and $D_2O$ solvent over Pd/C at different $H_2$ pressures and temperatures, following a kinetic analysis method developed by ref. [29]. (Table S3). The results are shown in Fig. S4 and the calculated thermodynamic parameters are summarized in Table 1. It is noteworthy that this method gives the same result (equilibrium constant ($K_{H_2}^o$)= 0.215 at pH 7 and room temperature) as our recently reported value measured via a different method using transient response of $D_2$ replacing adsorbed $H_2$ under equivalent conditions ($K_{H_2}^o$ = 0.20)[1], upholding the accuracy of these measurements. The present work shows that $K_{H_2}^o$ decreased with pH, i.e., from 0.215 at pH 7 to 0.09 at pH 1.6. The enthalpy change ($\Delta H^o$), calculated from the measured $K_{H_2}^o$ at different temperatures using the Van't Hoff equation, decreased from −42 kJ $mol_{H_2}^{-1}$ (pH 7) to −32 kJ $mol_{H_2}^{-1}$ (pH 1.6). On average, $\Delta H^o$ increased by 2 kJ $mol_{H_2}^{-1}$ per unit pH, which is close to the value of 2.7 kJ $mol_{H_2}^{-1}$ $pH^{-1}$ we calculated using the cyclic voltammetry data in Zheng et al.[4]. This suggests a less exothermic $H_2$ adsorption process on Pd under acidic conditions, in good agreement with the previous electrochemical analyses for platinum group metals (Pd, Pt, Rh, Ir)[3,4]. Considering the minor impact of pH on the adsorption of furfural, the decrease in the metal-H binding strength with increasing hydronium ion concentration is attributed to organization at the metal surface and in the Helmholtz layer[2,5].

In presence of two adsorbed species in this bimolecular reaction, it is important to address whether H and furfural compete for the same adsorption site. The kinetic analysis of H/D exchange between $H_2$ and $D_2O$ (Table S3 and ref. [29]) was performed in the presence and absence of furfural. In essence, we evaluate the impact of furfural on the adsorption rate of $H_2$ in transient state, which is the sum of desorption rates of $H_2$, HD, $D_2$ (and FAL formation rate in the case of furfural). Figure 5 shows that at both pH 1.6 and 5.8, the rate of $H_2$ adsorption on Pd decreases by ~50% in 30 mM furfural solution compared to its absence. Given that >10 mM is the saturation concentration (isotherms in Fig. 3a), we infer that the saturated adsorption of furfural leaves only half of the total metal sites accessible to $H_2$. It is reasonable that at low furfural concentrations, the oxygenate competes with $H_2$ for adsorption sites on Pd. Once furfural reaches, however, the state of saturation, the Pd surface cannot take more furfural molecules, but hydrogen atoms of smaller size, which can access Pd sites located at the vacancies between the tilted adsorbed furfural molecules. At this stage, $H_2$ adsorption and the resulting $\theta_H$ on the 50% remaining sites are independent of furfural adsorption and in consequence, the molecules do not compete for sorption. Such an argument agrees with the observation of 0th reaction order in furfural (Fig. S1b). Therefore, we conclude that $H_2$ adsorbs at sites not occupied by furfural and, hence, can be described by a non-competitive adsorption model in the kinetic analysis. In passing, we would like to note that our previous work reported that saturated benzaldehyde blocked 30% $H_2$ adsorption sites on Pd[23].

## Kinetic model of hydrogenation

Having established the mode and strength of adsorption of the reactants, a kinetic model is developed to quantify impact of the hydronium ion concentration. Table 2 compiles the elementary steps: (1) dissociative adsorption of $H_2$ on the active site to two H*; (2) adsorption of a furfural molecule; (3) the first H addition to the adsorbed furfural molecule (via the LH pathway and/or via PCET and the reverse Volmer step); and (4) the second H addition to the adsorbed furfural-H intermediate (via the LH pathway). The reaction order with respect to furfural concentration is zero, thus, Step 2 is quasi-equilibrated (Fig. S1b), whereas the 1st order in $H_2$ suggests the quasi-equilibrium established in Step 3 (Supplementary Note S4). Although our KIE study presents no evidence of the H addition pathway in Step 3 (Fig. 2), it remains sensible to have the constant $K_1^o$ accounting for the quasi-equilibrated surface species in our kinetic model because all species in this step are in the state of quasi-equilibrium regardless of pathways. Finally, Step 4 is the rds and it involves the addition of a surface H* atom to the carbonyl C of furfural, according to the KIE results (Fig. 2). We exclude the $H_2$ diffusion and FAL desorption from the kinetic model as justified in Note S2.

Considering the non-competitive adsorption of furfural and H in the kinetic regime of interest in this study, we derive the explicit expressions for the coverages of adsorbed hydrogen, furfural, and

**Table 1 | Thermodynamic parameters of $H_2$ adsorption on Pd/C ($H_2(g) + 2^* \rightarrow 2H^*$)[a]**

| pH | $\Delta H^o$ (kJ $mol_{H_2}^{-1}$) | $\Delta G^o$ (kJ $mol_{H_2}^{-1}$) @ 298 K | $\Delta S^o$ (J $mol_{H_2}^{-1}$ $K^{-1}$) | $K_{H_2}^o$ | | | |
|---|---|---|---|---|---|---|---|
| | | | | 298 K | 313 K | 323 K | 333 K |
| 1.6 | −31.5 | 6.0 | −125 | 0.090 | 0.050 | 0.034 | 0.025 |
| 3.0 | −35.0 | 5.3 | −135 | 0.119 | 0.062 | 0.039 | 0.027 |
| 4.5 | −37.3 | 5.1 | −142 | 0.128 | 0.061 | 0.038 | 0.029 |
| 5.8 | −40.3 | 4.1 | −149 | 0.189 | 0.091 | 0.057 | 0.036 |
| 7.0 | −42.3 | 3.8 | −154 | 0.215 | 0.105 | 0.056 | 0.039 |

[a]Calculated from the data set in Fig. S4.

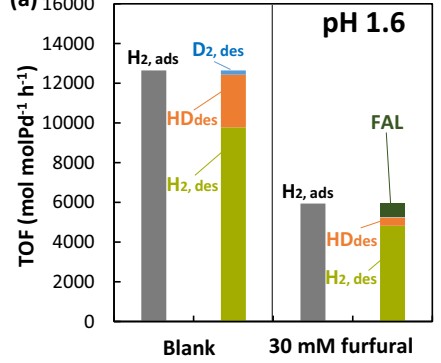
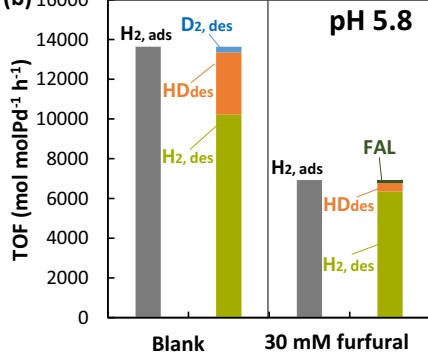

**Fig. 5 | $H_2$ adsorption on Pd/C in the presence of furfural.** Adsorption (ads) rate of $H_2$, desorption (des) rates of $H_2$, HD, and $D_2$, as well as FAL formation rate during the H/D exchange between $H_2$ gas (10 bar) and $D_2O$ solvent on 10 mg Pd/C catalyst in the presence of furfural at room temperature, **a** pH 1.6 and **b** pH 5.8. The rate calculations are performed based on the kinetic analysis summarized in Table S3[29]. Source data are provided as a Source Data file.

## Table 2 | Elementary steps in furfural hydrogenation

| (Step 1) H$_2$ adsorption | H$_2$ + 2 * $\rightleftharpoons$ 2 H* | $K^o_{H_2}$ |
|---|---|---|
| (Step 2) Furfural adsorption | F + * $\rightleftharpoons$ F* | $K^o_F$ |
| (Step 3) 1st H addition | F* + H* $\rightleftharpoons$ FH* + * | $K^o_1$ |
| | F* + H$^+$ + e$^-$ $\rightleftharpoons$ FH* | $K^o_{1,PCET}$ |
| | H* $\rightleftharpoons$ H$^+$ + e$^-$ + * | $K^o_{1,Volmer}$ |
| (Step 4) 2nd H addition | FH* + H* $\rightarrow$ FH$_2$ + 2 * | $k_2$ |

## Table 3 | Summary of the thermodynamic and kinetic parameters of furfural hydrogenation on Pd/C at 0.25–10 bar H$_2$ and room temperature

| pH | $K_F^{o\,a}$ [$G_F^o$ (kJ mol$^{-1}$)] | $K_{H_2}^{o,\,b}$ [$G_{H_2}^o$ (kJ mol$^{-1}$)] | $K_1^{o\,c}$ [$G_1^o$ (kJ mol$^{-1}$)] | $k_2$(s$^{-1}$)$^c$ |
|---|---|---|---|---|
| 1.6 | 2297 ± 282 [−19.2 ± 0.3] | 9.03×10$^{-2}$ [5.97] | 10.3×10$^{-4}$ [16.4] | 1322 |
| 3 | | 11.9×10$^{-2}$ [5.27] | 8.52×10$^{-4}$ [17.0] | 1127 |
| 5.8 | | 18.9×10$^{-2}$ [4.13] | 4.43×10$^{-4}$ [18.5] | 616 |
| 7 | | 21.5×10$^{-2}$ [3.80] | 2.56×10$^{-4}$ [19.8] | 357 |

The corresponding standard Gibbs free energy (**G°**) is given in square brackets.
$^a$Obtained from regression of furfural adsorption isotherms with the Langmuir model.
$^b$Measured by kinetic analysis of H/D exchange between H$_2$ and D$_2$O.
$^c$Acquired by furfural hydrogenation kinetic model fitting.

intermediate ($\theta_H$, $\theta_F$, $\theta_{FH}$) (Eq. S5). These expressions are used to derive the rate equation in terms of measurable equilibrium constants and experimental conditions (Eq. (2)), which can be further simplified (Eq. (3)). The details of equation derivation are provided in Supplementary Note S2.

$$r = k_2 \theta_H \theta_{FH} = \frac{k_2 K_{H_2} P_{H_2} K_1 K_F C_F}{\left(K_{H_2}^{0.5} P_{H_2}^{0.5} + 1\right)\left(K_F C_F + K_1 K_F C_F K_{H_2}^{0.5} P_{H_2}^{0.5} + 1\right)} \quad (2)$$

$$r \approx k_2 K_{H_2} P_{H_2} K_1 \quad (3)$$

where $P_{H_2}$ and $C_F$ are the H$_2$ pressure and furfural concentration, respectively. It is noteworthy that the derived rate equation (Eq. (3)) is consistent with the measured reaction orders in H$_2$ and furfural (Fig. S1).

Regression analysis is then performed by fitting the reaction data and the corresponding conditions as well as the measured adsorption equilibrium constants ($K_F^o$, $K_{H_2}^o$) into Eq. (2). To assure the data quality, we verify if the Pd/C catalyst surface is altered during the catalytic run. The recycling test shows that the activity of the recovered catalyst remains the same as in cycle 1 (Fig. S5). The spent catalysts were subjected to characterization. The XPS results show that the catalyst surface was dominated by Pd$^0$ with a small amount of Pd$^{2+}$ (Fig. S6c). The small amount of Pd$^{2+}$ was likely from oxidation by exposure to air during catalyst recovery and storage prior to XPS analysis. In the XRD patterns, diffraction peaks of metallic Pd were predominant, while oxidized Pd was not observed (Fig. S7). Using the Scherrer equation, the Pd particle size was determined to be 4.6 ± 0.3 and 4.5 ± 0.3 nm before and after the reaction. The evidence point to the unchanged surface of Pd/C and, thus, substantiates the validity of the reaction data for the regression analysis (Eq. (2)). The parity plots show that the computed TOFs of hydrogenation agree with the experimental data very well (Fig. S8). The key inputs for and outputs from the regression are summarized in Table 3. As it is changed from pH 7 to pH 1.6, the equilibrium constant for the first H addition (Step 3; $K_1^o$) is found to increase from 2.56 × 10$^{-4}$ to 10.3 × 10$^{-4}$, corresponding to the Gibbs free energy change of 20 and 16 kJ mol$^{-1}$, respectively. This implies that the

1$^{st}$ H addition is thermodynamically more favorable at high hydronium ion concentrations.

For parameters in the rate equation (Eq. (2)), the rate constant ($k_2$) of furfural hydrogenation increased from 357 s$^{-1}$ at pH 7 to 1322 s$^{-1}$ at pH 1.6 (Table 3 and Fig. 6a), suggesting a smaller intrinsic activation free energy at low pH. The coverages $\theta_H$ and $\theta_{FH}$ (calculated using Eq. (S5)) behave in an opposite manner. At 1 bar H$_2$ for example, the $\theta_H$ decreases by -27% when changing from pH 7 to pH 1.6, whereas $\theta_{FH}$ increases by 160% (Fig. 6a). Note that the surface coverage of furfural remains high at ≥0.9 under all conditions, which agrees with the saturation region in the furfural adsorption isotherms (Fig. 3a). We insert the values of regressed coverages into the derived expressions of reaction orders (Eqs. S8 and S10), which are then estimated to be 0.84 ± 0.08 in H$_2$ and 0.014 ± 0.001 in furfural at 0.25−10 bar H$_2$. The outputs are close to the experimental measurements (Fig. S1), validating our kinetic model. In summary, the overall increase in the hydrogenation rate is the multiplication of changes in $k_2$ (+270%), $\theta_{FH}$ (+160%), and $\theta_H$ (−27%).

We show, hence, that pH has the most significant impact on the rate constant of the rds (i.e., $k_2$), implying that the intrinsic activation energy ($E_a$) is affected the most. At a lower pH, H* assumes a higher adsorbed state as the hydrogen binding strength is weakened by the surrounding hydronium ions of high concentration ($\Delta H^o$ measurement for H$_2$ adsorption; Table 1). This reduces the energy level difference between the adsorbed state and transition state, resulting in a smaller intrinsic $E_a$ at low pH, i.e., 41.4 ± 6.3 kJ mol$^{-1}$ at pH 1.6 vs. 54.1 ± 6.2 kJ mol$^{-1}$ at pH 5.8 (Fig. 7). Thus, the change in HBE dominates the impact in the pH dependence of hydrogenation kinetics. The H atoms have a higher excess chemical potential to attack furfural at low pH. Note in passing that we reported similar phenomena as the pH-dependent HBE governs the Pt-catalyzed hydrogenation of phenol[2]. In comparison, the change in furfural adsorption with pH is insignificant.

We observe also that the two equilibrium constants, $K_1^o$ and $K_{H_2}^o$ are inversely correlated (Fig. 6b). Thus, the weaker binding of hydrogen leads to a higher equilibrium constant for the half-hydrogenated state of furfural, i.e., the chemical bonding of half-hydrogenated furfural is thermodynamically preferred over adsorption of atomic hydrogen as the latter bonding is weakened by the higher hydronium ion concentration.

The mechanistic and kinetic analyses show that the rate-determining step, i.e., the addition of the second H atom, in the hydrogenation of furfural on Pd in aqueous phase follows a Langmuir−Hinshelwood mechanism and does not occur via a proton-coupled electron transfer. Higher concentrations of hydronium ions increase the rate, while they do not change the mode of hydrogen addition. The first-order dependence in H$_2$ and isotope labeling show that the addition of H* to the half-hydrogenated intermediate is the rate-determining step.

Adsorption measurements suggest that the heat of H$_2$ dissociative adsorption decreased with increasing hydronium ion concentration, while furfural adsorption has been concluded not to be influenced. Although furfural at saturation blocks half of the metal sites (concluded from H/D exchange experiments), the H$_2$ adsorption on the remaining sites is not impeded by furfural once it has reached the saturation level, i.e., pointing to a noncompetitive sorption.

The analysis of the kinetic data shows that acidity enhances the hydrogenation rate by increasing the rate constant ($k_2$) and the coverage of the half-hydrogenated intermediate ($\theta_{FH}$), although the improvement is slightly counteracted by the decreased hydrogen coverage ($\theta_H$). Combined, the changes with hydronium ion concentrations lead to an eightfold increase in the furfural hydrogenation rate. The rate constant $k_2$ representing the addition of the second H atom is more pH-sensitive than the coverages and is considered to be the major contributor to the pH dependence. The present findings confirm that weaker hydrogen binding−in this case induced by the

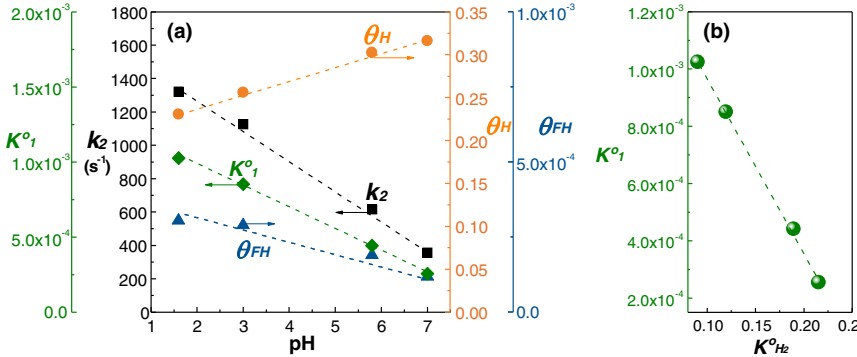

**Fig. 6 | Equilibrium constants, rate constant, and coverages for Pd-catalyzed furfural hydrogenation. a** Calculated $K_1^o$, $k_2$, $\theta_{FH}$ and $\theta_H$ as a function of pH in furfural hydrogenation at 1 bar $H_2$. **b** Correlation between $K_1^o$ and $K_{H_2}^o$. The coverages are calculated using Eq. S5 with the measured and computed equilibrium constants as well as experimental reaction rates and the corresponding conditions. Source data are provided as a Source Data file.

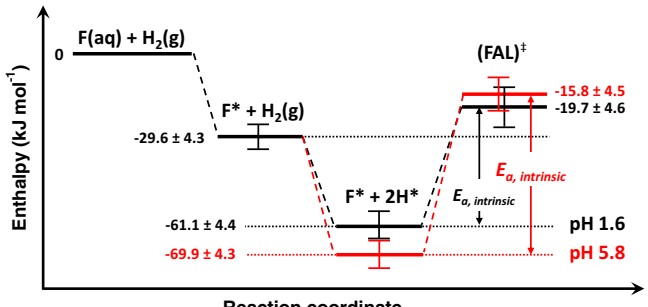

**Fig. 7 | Energy diagram for the illustration of pH effect on hydrogenation kinetics.** The profile in black color represents the reaction at pH 1.6, and the red one represents that at pH 5.8.

increase in hydronium ion activities—leads to higher rate constant of hydrogen addition. The adsorbed state of the reacting substrate is hardly influenced and, thus, does not impact kinetics. However, its adsorption geometry is significant, as the tilted adsorbed furfural molecules favor H* addition to -CHO rather than the unsaturated furan ring, yielding alcohol product of high selectivity.

It is hypothesized that such pH dependence can be generalized for hydrogen addition reactions and is primarily caused by the increase in the excess chemical potential of adsorbed hydrogen. If further examples with a broader range of substrates and catalysts can be substantiated, it would evolve as major design parameter for hydrogenation catalysts. The present work advances this path by showing the specific pH effects on kinetic and thermodynamic parameters, providing fundamental insights to advance aqueous-phase catalytic systems of high efficiency.

## Methods

### Chemicals

Furfural, FAL, and THFAL were purchased from Sigma Aldrich. They were used as substrates in reactions and/or as standard compounds for calibration of analytical instrument. Phosphoric acid (>85 wt%) and monobasic and dibasic sodium phosphate (Sigma Aldrich) were used for the preparation of aqueous buffer solutions (pH 1.6–7). The catalyst used was Pd (5 wt%) supported on activated carbon (Pd/C; Sigma Aldrich), with the metal dispersion of 32% based on $H_2$ chemisorption analysis. Note that Pd is a prospective catalyst to achieve high-performance hydrogenation and it forms the foundation of cost-effective catalyst design, such as the recent advance in the development of Pd single-atom catalysts and Pd-based alloy catalysts[30–32].

Carbon support was chosen in this study because it is more chemically inert in aqueous solution compared to other common supports (e.g., zeolites, alumina), which minimizes the interference by any other possible active sites on the support itself. All chemicals were used as received.

### Catalytic hydrogenation

Before the hydrogenation experiment, Pd/C catalyst (10 mg) was treated in the phosphate buffer (30 mL) at 30 bar $H_2$ and room temperature for 30 min under stirring (600 rpm) in a stainless-steel batch reactor (PARR, 100 mL capacity). Note that PdO reduction is feasible at room temperature according to temperature programmed reduction studies in the literature[33,34], and also shown by X-ray photoelectron spectroscopy (XPS) analysis of our fresh and pretreated Pd/C catalysts (Fig. S6a, b). After the in-situ Pd/C pretreatment, furfural was added to the reactor to reach a concentration of 30 mM unless otherwise specified, followed by $N_2$ purging to remove air. The reactor was heated to the operating temperature (25–100 °C) and then charged with $H_2$ (0.25–10 bar) to initiate the reaction. Stirring was maintained at 600 rpm throughout the process. Samples were taken from the batch at certain time intervals for product analysis. In the recycling test, the separated catalyst was rinsed with water, oven-dried, and directly used in the second cycle.

Aqueous samples from the batch were subjected to neutralization using NaOH and extraction in ethyl acetate at a solvent-to-sample ratio of 2:1 (v/v). The extraction solvent contained diphenyl ether (5 mM) as an external standard. For substrate and product quantification the organic phase was analyzed by gas chromatography with flame ionization detection (GC/FID; Hitachi). Turnover frequency (TOF) was calculated based on the formation of FAL (Eq. (4)). Selected samples were analyzed by GC-mass spectrometry (GC/MS; Agilent) for tracing isotopes.

$$TOF = \frac{FAL\,(mmol)}{Pd/C\,(g) \times Pd\,loading\,(wt\%) \times Pd\,dispersion\,(\%) \times reaction\,time\,(h)}$$

(4)

where Pd loading is 5 wt% and Pd dispersion is determined to be 32% by $H_2$ chemisorption.

### Furfural adsorption

To measure furfural adsorption isotherms at varying pH, Pd/C (150 mg) was suspended in phosphate buffer solutions that contained furfural at different initial concentrations under stirring in an $N_2$ atmosphere. The equilibrium concentration was determined by GC/FID as described above. The adsorption isotherms were fitted into

the Langmuir adsorption model:

$$q = \frac{QK_F^o C_{F,eqm}}{1 + K_F^o C_{F,eqm}} \qquad (5)$$

where $q$ is the adsorbed amount of furfural on Pd/C, $Q$ is the maximum adsorbed amount of furfural, $K_F^o$ is the equilibrium constant for furfural adsorption, and $C_{F,eqm}$ is the equilibrium concentration of furfural. Then, the Gibbs free energy ($\Delta G^o$) can be calculated:

$$\Delta G^o = -RT\ln K_F^o \qquad (6)$$

where $R$ is the gas constant (8.3145 J mol$^{-1}$K$^{-1}$) and $T$ is the temperature (K). The heat of adsorption of furfural was measured using a Setaram Calvet C80 calorimeter.

## Hydrogen adsorption

The equilibrium constants for hydrogen adsorption ($K_{H_2}^o$) at different pH values were determined by measuring the rates of adsorption and desorption through kinetic analysis of $D_2O$ reacting with $H_2$ to HDO, HD, and $D_2$[29]. In brief, Pd/C catalyst (100 mg) immersed in $D_2O$ (30 mL) was exposed to varying $H_2$ pressures in the stainless steel autoclave same as in hydrogenation experiments (PARR, 100 mL capacity). Then, HD (g) and $D_2$ (g) resulting from H/D exchange reactions were monitored using a mass spectrometer. The measured formation rates were used for graph plotting as shown in Table S3 and $K_{H_2}$ was calculated as the square of the slope in the plot (Fig. S4).

## Data availability

All data generated in this study are provided in the Supplementary Information/Source Data file. Source data are provided with this paper.

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

## Acknowledgements

I.K.M.Y. is grateful to the Alexander von Humboldt Foundation for the Research Fellowship. J.A.L. acknowledges the support by the U.S. Department of Energy (DOE), Office of Science, Office of Basic Energy Sciences (BES), Division of Chemical Sciences, Geosciences and Biosciences (Impact of catalytically active centers and their environment on rates and thermodynamic states along reaction paths, FWP 47319). Y.L. and W.Z. acknowledge the support of the Open Project Program of Academician and Expert Workstation, Shanghai Curui Low-Carbon Energy Technology Co., Ltd.

## Author contributions

I.K.M.Y., W.Z., and J.A.L. conceived the research; I.K.M.Y. performed the catalytic reaction; F.D. and W.Z. characterized the catalysts; X.C. and G.C. measured the $H_2$ adsorption; I.K.M.Y., Y.L., and Z.W. analyzed reaction kinetic and thermodynamic results. The manuscript was written through the contributions of all authors. All authors have given approval for the final version of the manuscript.

## Funding

## Competing interests

The authors declare no competing interests.
