## [Peer Review File · Nature Communications]

Impact of hydronium ions on the Pd-catalyzed furfural hydrogenationREVIEWER COMMENTS

Reviewer #1 (Remarks to the Author):

Overall this is a well written and interesting manuscript. The authors have studied the mechanism of furfural hydrogenation in aqueous media and discussed the impact of hydronium ions on the reaction mechanism. The kinetic and thermodynamic study has been performed nicely and the conclusions derived are indeed interesting. However, various points would need to be addressed before the manuscript is published (some of these points are easy to address and some others may require presenting some additional evidence).

Specific comments:

1. One thing that is not sufficiently explained in the introduction is the reason behind the choice of Furfural as the target molecule. The authors have one sentence stating that Furfural is an important molecule in a carbon neutral future...". I believe the authors could very briefly explain this statement. Why furfural is an important biomass derived target molecule and more importantly why furfuryl alcohol is an important product too. I think this will add value to the manuscript.
2. The manuscript would also benefit with a statement about the choice of catalyst (Pd/C). Pd based catalysts have been utilised in the literature for the hydrogenation of Furfural ranging from Pd supported nanoparticles to Pd based Single Atom Alloys. I believe the authors should make a relevant comment about this. More critically however, the authors should explain the reason behind the choice of the carbon support. One of the most common problems with Pd heterogeneous catalysts is the carbon contamination of the Pd particles during the reaction. Typically, the Pd catalysts are reactivated using oxidation / hydrogenation treatments. Therefore, from a more practical point of view carbon would not be an ideal support if the catalyst needs reactivation. I think the authors could add some relevant comments to address this.
3. Selectivity of the reaction: the authors do not discuss the selectivity of the reaction. I see in the conclusions that there is a statement "...yielding alcohol product of high selectivity". How much selective is the reaction towards furfuryl alcohol? Do the authors imply that the selectivity is 100%. I believe this is important and must be clear in the manuscript. The literature suggests that in both aqueous and organic media other products are possible.
4. Carbon balance: I think the authors should discuss the carbon balance of the reaction. This becomes particularly important as one would expect that certain amount of furfural will decompose and leave some organic deposits on the Pd surface.
5. Have the authors tried to reuse the catalysts for a second run?
6. The TOFs reported by the authors correspond to the TOFS of furfuryl alcohol production and they are based on equation 3 given in the methods section. I think it would be important to also report in the manuscript the true TOFs of the reaction (Number of furfural molecules reacted)/ (Number of sites) x

(time). This is crucial as the authors would correct for the number of sites and not the amount (mmol) of Pd used. I think the authors should comment on this in the paper or the supplementary information.

7. I note that the authors have used an Aldrich Pd/C catalyst without any characterization. It would be interesting to see some basic characterization of the material before and after reaction. More specifically the XRD, and TEM are important to have an idea of particle size distribution and dispersion and how these change upon reaction. I think the Pd active surface area is also important to be measured (or estimated). This would help derive some TOFs with respect to the active sites.

8. What %Conversion have the authors achieved with the specific way they have performed the reaction.

9. The hydrogenation of Furfuryl alcohol is sensitive on the electronic and geometric characteristics of the catalytic system. The authors, in order to maintain the pH they have used sodium phosphate. I guess that the phosphate (and the sodium) will also adsorb on the catalyst surface, and they can act as poisons or as promoters or simply be spectators during the reaction. Have the authors accounted for this?

10. With respect to the previous comment have the authors performed any post catalyst characterization (possibly XPS) to see what is on the surface of the material.

Reviewer #2 (Remarks to the Author):

This report compares the validity of two reaction mechanisms for the hydrogenation of furfural to furfuryl alcohol. The first is a classical Langmuir-Hinshelwood (LH) mechanism based on hydrogen dissociation on the surface of Pd nanoparticles, and stepwise addition of adsorbed hydrogen to adsorbed furfural. The second is a proton-coupled electron transfer (PCET) mechanism.

The second is discarded early on because increases in hydronium concentration by 100,000 times, only increase the reaction rate by ~10 times. This is inconsistent with the kinetics of PCET.

In the second part of the report, the LH mechanism is evaluated in detail to identify the most important surface species, the rate-determining step, and the effect of pH on the hydrogen binding energy over Pd. Understanding this effect and the heat of adsorption of furfural on Pd, are key to explaining why reaction rate increases by lowering pH.

I think the conclusions of the report are solid and based on new information and kinetic analysis that are original to this report.

I am not strongly enthusiastic about recommending this report for Nature Communications because the approach to solving this mechanistic challenge is very classical; very well done and detailed, but not new in itself. The reaction itself is not particularly important; furfuryl alcohol has been produced without problems from furfural for many decades. And in heterogeneous catalysis extrapolation to other organic species is an educated guess, at best.

On the other hand, electrochemical processes and heterogeneous catalysis often occur simultaneously, and the individual contributions to observed reactivity can be disentangled by following variations of the research plan implemented in this report. In this sense, this report will be useful to the heterogeneous catalysis community as a blueprint to answer similar or related questions about catalysis and the metal-aqueous interface in the future.

The report could be published as is, but I found both the abstract and the introduction hard to follow. They deserve to be edited for clarity. In the first, the sentences do not follow each other logically and the report's key findings are lost. In the second, the paragraphs do not follow each other very logically (consider starting with the last one). I found the rest of the report clear and concise.

Reviewer #3 (Remarks to the Author):

Review of NCOMMS-22-16564

The article » Impact of hydronium ions on the Pd-catalyzed furfural hydrogenation « by Yu et al. presents a mechanism study of furfural aldehyde hydrogenation in aqueous media. Sequential addition of hydrogen atoms by Langmuir-Hinshelwood mechanism is proposed and confirmed experimentally with variation of pH and isotope labeling, as well as in silico. This paper contradicts the proton-coupled electron transfer (PCET) mechanism recently published in Nature Catalysis. The methodology is rather standard, while it serves its purpose well. Topic is interesting for the catalytic community, however its relevance for the Nature Communication audience too specific and the impact of the research is rather limited, maybe a publication in Nature Catalysis will fit more, however this is a ultimate authority of the editor to decide. Serious flaw of the current article are suspiciously low activation barriers reported for diffusion-controlled reactions in water solution. Should the editor find this paper sufficiently relevant for wide audience of Nature Communication, I would consider this article suitable for publishing after a major revision. Careful addressing of comment 5 on the activation barriers will be crucial for the eventual acceptance.

1. On page 12, it is claimed that only 50% of remaining sites are available for hydrogen adsorption due to the fact that furfural is only blocking these sites and are independent of furfural adsorption and do not compete for the same sites. This seems to be an overzealous assumption to assume that they are non-competitive as there are only metallic Pd sites available on the Pd/C catalyst. The presented model also only assumes that 1 molecule of furfural binds to 1 site which is not necessarily true.

2. Why were the Pd catalysts pretreated in H₂ at room temperature? How can you assure that all of the Pd metallic species were reduced? This may also affect the total number of sites that were considered available for furfural and/or H₂ adsorption. With no TPR or XPS data presented for the Pd/C catalyst, it is possible that there are different amounts of reduced and oxide sites available at each of your reaction temperatures between 25 – 100 °C after reduction in situ.

3. Does hydrogen partial pressure really affect the equilibrium coverage according to $n=0.5$? H₂-TPD (with pulse-chemisorption) experiment could demonstrate the validity of Eqn. S1a.

4. It should be demonstrated and proven that reactor operates in kinetic regime. No effect of external and internal mass transfer limitations on the kinetic rate(s) should be determined either experimentally, via kinetic modeling or other criteria (e.g. Weisz–Prater).

5. Page 7: Apparent activation barriers of 9.9 and 13.8 kJ mol⁻¹ are very suspicious. In water solutions, diffusion-controlled reactions proceed with approx. 10^9 s⁻¹, which is the highest reaction rate for a reaction in water solution. The corresponding diffusion barrier is 21.8 kJ mol⁻¹ from Eyring-Polanyi equation. This barrier is determined from the mobility of water molecules (water diffusion coefficient). The only exceptions are reactions of proton hopping between two adjacent water molecules or water molecule chains (Grotthuss mechanism of “proton wire”). Basically, all reactions with intrinsic barriers lower than 21.8 kJ mol⁻¹ will be diffusion controlled. If the authors claim that their barriers are lower, they need to justify this mechanistically. Even arguing that furfural is hydrogenated on the surface, which is true, does not circumvent this problem since furfural *must* be delivered to the surface through water, hence being subject to the diffusion barriers.

6. Page 7: The barrier is not the energy level difference between the transition state and the ground state because these are two different things. Transition states are saddles on the potential energy surface, which the reaction starting at the initial state must overcome. The ground state means that there is no electron excitation, as opposed to excited state.

7. Page 8: In Ref [21], -23 kJ mol⁻¹ is the free energy of adsorption of FAL and 2 H* and not only FAL is the authors claim. This means that they cannot compare the value -19 kJ mol⁻¹ directly. Moreover, Gibbs free energy is *a strong* function of temperature, which the authors do not state. While the Gibbs free energy can be as low as -19 kJ mol⁻¹, the temperature-independent adsorption energy (or enthalpy) must be stronger (=more negative).

8. Page 8: The authors state the furfural-Pd bonds are strong and remain unaffected by H₂O-involving interaction. The very same Ref [21] contradicts this on Page 18 of the supplementary by stating: »Furfural adsorption energy in the gas phase was taken from experimental TPD values of -0.98 ± 0.20 eV[16,17].« This is a change from 100 kJ mol⁻¹ to 19 kJ mol⁻¹, which is not *unaffected*.

9. In fact, the authors admit this on Page 9: »Without the influence of water (i.e., in ultrahigh vacuum (UHV)), the furfural adsorption enthalpy (ΔH_{fadsg}) was computed to be about -164 kJ mol⁻¹ (-1.7 eV) if furfural lies flat on a clean Pd surface at a low density of 0.25 monolayer (ML) [14].« and »The enthalpy changes to 179 about -97 kJ mol⁻¹ as the surface becomes crowded at 0.5-1 ML and furfural binds to Pd via the – CHO group, resulting in a tilted conformation.[14]«

10. Page 9: I am not sure that talking about one side of the molecule being solvated makes sense. Solvation (or adsorption on Pd) is a complex process that entails significant electron density redistribution.

11. Page 9: Why is one fourth of furfural area blocked by adjacent furfural molecules? Provide justification.

$$k_{cat} = \frac{k_B T}{h} e^{-\frac{\Delta G^\ddagger}{k_B T}}$$

Reviewer #4 (Remarks to the Author):

The mechanism of the selective hydrogenation of polar carbonyl groups in aqueous phase and the effect of pH on the reaction rate are intensely debated. Several groups have proposed water-mediated hydrogenation steps to be involved. Using detailed kinetic and isotope measurements, the authors claim to show “unequivocally” that the hydrogenation occurs via sequential addition of hydrogen atoms via a Langmuir Hinshelwood mechanism. These are carefully executed experiments, but the evidence is unfortunately not sufficient to convince me.

The key experiment is shown in Figures 2 and 3. When H₂ is replaced D₂, the TOF decreases significantly and FAL product mass spectra also shift up. While the H-atom that hydrogenates the C atom in RCHO is generally accepted to originate from adsorbed H, the debate is about the H atom that attacks the O atom. Unfortunately, FAL alcohol rapidly exchanges its proton with water (Figure 3a) and the origin of that H cannot be determined with isotope experiments. The increase in m/z 100 from D₂/H₂O to D₂/D₂O actually suggests that this “H” comes from water.

Hydrogen adsorption is not kinetically relevant, as the authors indicate and is suggested by the first order kinetics in H₂. Figure 10 shows that the kinetically relevant energy difference is between F* + H₂(g) and the (FAL) transition state. The energy diagram in Figure 10 is somewhat unexpected since weaker H* adsorption (and hence a less favorable hydrogenation reaction step since the product FAL(l) does not change) corresponds with a more stable transition state and a lower activation energy. A BEP relation for a typical LHHW elementary step would suggest the opposite trend, ie a higher activation energy for a less favorable reaction step.

The argument for tilted F adsorption via the RCHO group are based on the thermodynamic cycles in Figure 6. These calculations are based on experimental datapoints and several assumptions and not fully conclusive. Spectroscopic data could provide more direct proof for the geometry. According to the thermodynamic cycle in Figure 6, parallel adsorption should be more favorable, but might not occur at high coverage. From these arguments, one would expect a change in selectivity and in heat of adsorption for very low F concentrations, but this is not observed.

Kinetically, it is very difficult to distinguish competitive and non-competitive adsorption. Since F adsorbs stronger than H₂, it is expected that the F coverage is saturated and H₂ adsorbs at the available sites between F molecules. At very low F concentrations, one would however expect to see kinetic effects of competition between F and H₂.

Reviewer #1 (Remarks to the Author):

Overall this is a well written and interesting manuscript. The authors have studied the mechanism of furfural hydrogenation in aqueous media and discussed the impact of hydronium ions on the reaction mechanism. The kinetic and thermodynamic study has been performed nicely and the conclusions derived are indeed interesting. However, various points would need to be addressed before the manuscript is published (some of these points are easy to address and some others may require presenting some additional evidence).

Response:

Thank you for your time and effort in reviewing the manuscript and providing constructive suggestions. We make clarifications in this revision and performed supplementary experiments for new evidence. Please see below our point-to-point responses to your comments.

Specific comments:

1. One thing that is not sufficiently explained in the introduction is the reason behind the choice of Furfural as the target molecule. The authors have one sentence stating that Furfural is an important molecule in a carbon neutral future...". I believe the authors could very briefly explain this statement. Why furfural is an important biomass derived target molecule and more importantly why furfuryl alcohol is an important product too. I think this will add value to the manuscript.

Response:

The importance of furfural and furfuryl alcohol is elaborated in the revised manuscript (Page 2).

"Furfural is chosen as a model compound in this study to gain fundamental insights for such more complex reactants. It can be derived from lignocellulosic biomass as an abundant renewable resource, drawing significant attention as a versatile platform molecule to serve wide applications in a carbon neutral future.⁸⁻¹³ Furfuryl alcohol (FAL) is one of the important hydrogenation products of furfural and can be used as the monomer for producing FAL resin.¹⁴ In addition, FAL can be subjected to upgrading processing to product methyl furan as a fuel additive and levulinic acid as a precursor of green solvents, fuels, polymers, etc.^{15,16"}

2. The manuscript would also benefit with a statement about the choice of catalyst (Pd/C). Pd based catalysts have been utilised in the literature for the hydrogenation of Furfural ranging from Pd supported nanoparticles to Pd based Single Atom Alloys. I believe the authors should make a relevant comment about this. More critically however, the authors should explain the reason behind the choice of the carbon support. One of the most common problems with Pd heterogeneous catalysts is the carbon contamination of the Pd particles during the reaction. Typically, the Pd catalysts are reactivated using oxidation / hydrogenation treatments. Therefore, from a more practical point of view carbon would not be an ideal support if the catalyst needs reactivation. I think the authors could add some relevant comments to address this.

Response: As suggested, we added a comment on the use of different Pd catalysts in the literature. Carbon support was chosen in this study because it is more chemically inert in aqueous phase compared to other common supports (e.g., zeolites, alumina), which minimizes the interference by any other possible active sites on the support itself. The justification is added in Page 19.

“Note that Pd remains a prospective catalyst to achieve high-performance hydrogenation and it forms the foundation of cost-effective catalyst design, such as the recent advance in the development of Pd single-atom catalysts and Pd-based alloy catalysts.³⁰⁻³² Carbon support was chosen in this study because it is more chemically inert in aqueous solution compared to other common supports (e.g., zeolites, alumina), which minimizes the interference by any other possible active sites on the support itself.”

3. Selectivity of the reaction: the authors do not discuss the selectivity of the reaction. I see in the conclusions that there is a statement “...yielding alcohol product of high selectivity”. How much selective is the reaction towards furfuryl alcohol? Do the authors imply that the selectivity is 100%. I believe this is important and must be clear in the manuscript. The literature suggests that in both aqueous and organic media other products are possible.

Response: We added a table in Supporting Information to clarify the product selectivity in the revised manuscript. A copy is below.

Table S1. Selectivity of furfuryl alcohol (FAL), tetrahydrofurfural (THFF), and tetrahydrofurfuryl alcohol (THFAL) as well as carbon balance at furfural conversions of around 20-35%.*

pH	 Furfural conversion (mol%)	Product selectivity (mol%)			C balance (mol%)
		 FAL	 THFF	 THFAL	
1.6	19.4	88.8	3.6	1.5	98.8
3.0	21.0	92.1	5.1	3.5	100.2
4.5	29.7	94.5	5.3	4.9	101.4
5.8	35.5	83.9	7.4	7.6	99.6
7.0	22.6	81.9	11.9	6.2	100.0

* The reaction was performed with 30 mM furfural in 0.1 M phosphate buffer solution with 10 mg Pd/C at 1 bar H₂ and room temperature.

Selectivity = Yield / Conversion.

C balance = (Concentrations of products and unconverted Furfural) / Concentration of initial Furfural)

4. Carbon balance: I think the authors should discuss the carbon balance of the reaction. This becomes particularly important as one would expect that certain amount of furfural will decompose and leave some organic deposits on the Pd surface.

Response: The carbon balance is close to 100% as shown in Table S1 above, suggesting only very small amount of organic deposits are formed.

5. Have the authors tried to reuse the catalysts for a second run?

Response: We performed the recycling test, in which the reused catalyst retained the catalytic activity in the second run (Figure S4, a copy below). It showed the same catalytic activity as the fresh catalyst. The testing procedure was provided in the Methods Section of the revised manuscript.

Figure S4. Turnover number of furfural hydrogenation in the recycling test. The reaction was performed with 30 mM furfural in 0.1 M phosphate buffer solution (pH 1.6) with 20 mg fresh Pd/C in cycle 1 and 10 mg recovered Pd/C in cycle 2 at 1 bar H₂ and room temperature.

6. The TOFs reported by the authors correspond to the TOFS of furfuryl alcohol production and they are based on equation 3 given in the methods section. I think it would be important to also report in the manuscript the true TOFs of the reaction (Number of furfural molecules reacted)/ (Number of sites) x (time). This is crucial as the authors would correct for the number of sites and not the amount (mmol) of Pd used. I think the authors should comment on this in the paper or the supplementary information.

Response: The TOFs reported in this manuscript are based on number of exposed Pd sites, which was determined by H₂ chemisorption. We revised Eqn 3 (Page 20) as below for clarity:

$$TOF = \frac{FAL(mm\text{ol})}{Pd/C (g) \times Pd \text{ loading (wt\%)} \times Pd \text{ dispersion (\%)} \times reaction \text{ time (h)}} \quad (\text{Eqn. 3})$$

where Pd loading is 5 wt% and Pd dispersion is determined to be 32% by H₂ chemisorption.

7. I note that the authors have used an Aldrich Pd/C catalyst without any characterization. It would be interesting to see some basic characterization of the material before and after reaction. More specifically the XRD, and TEM are important to have an idea of particle size distribution and dispersion and how these change upon reaction. I think the Pd active surface area is also important to be measured (or estimated). This would help derive some TOFs with respect to the active sites.

Response: As suggested, we compared the XRD pattern of fresh and used catalysts, which resembles each other (Figure S6, a copy below). Diffraction peaks of metallic Pd were predominant, while oxides were not observed. Using Scherrer equation, the Pd particle size were determined to be 4.6 ± 0.3 and 4.5 ± 0.3 nm before and after reaction. The dispersion of Pd was measured to be 32% by H₂ chemisorption, and this value was used in the calculation of TOF.

Figure S6. XRD patterns of in-situ reduced Pd/C before and after catalytic furfural hydrogenation. In situ reduction condition: 0.1 M phosphate buffer solution (pH 4.5), 30 bar H₂ and room temperature for 30 min. Reaction condition: 30 mM furfural in 0.1 M phosphate buffer solution (pH 4.5) at 1 bar H₂ and room temperature for 1h

8. What %Conversion have the authors achieved with the specific way they have performed the reaction.

Response:

The reaction was carried out at various reaction times and conversions. As shown in the figure below, conversion exhibits strong linearity ($R^2 > 0.99$) even up to high conversions of 70%. TOF and rate are calculated based on the slope of conv. vs time.

Figure 1a. Furfural conversion as a function of reaction time. The reaction was performed with 30 mM furfural in 0.1 M phosphate buffer solution with 10 mg Pd/C at 1 bar H₂ and room temperature.

9. The hydrogenation of Furfuryl alcohol is sensitive on the electronic and geometric characteristics of the catalytic system. The authors, in order to maintain the pH they have used sodium phosphate. I guess that the phosphate (and the sodium) will also adsorb on the catalyst surface, and they can act as poisons or as promoters or simply be spectators during the reaction. Have the authors accounted for this?

Response: While phosphate anions are not attracted to the negatively charged surface of Pd (negative open circuit potential; Figure 1b), the hydration of sodium in water prevents the cations from adsorption. Besides, liquid calorimetry indicates very similar standard enthalpy for furfural adsorption ($\Delta H_{F_{ads,aq}}^0$) in phosphate buffer solutions ($\sim 30 \text{ kJ mol}^{-1}$; pH 3 and pH 5.8) and pure water (-31 kJ mol^{-1}), suggesting that the adsorption of phosphate and sodium on Pd are negligible. Therefore, we did not observe impacts associated with these ions.

10. With respect to the previous comment have the authors performed any post catalyst characterization (possibly XPS) to see what is on the surface of the material.

Response: The XPS results (Figure S5) and the corresponding discussion are added to the revised manuscript (Page 14). It shows that the catalyst surface is dominated by Pd⁰ with a small amount of Pd²⁺ (Figure S5b&c). The small amount of Pd²⁺ is likely from oxidation by exposure to air during catalyst recovery and storage prior to XPS analysis.

XRD characterization of Pd/C before and after reactions were also compared. Details are in the response to Comment 7.

Figure S5. XPS curves and fittings of (a) as-received Pd/C, (b) Pd/C after *in-situ* aqueous-phase reduction at 30 bar H₂ and room temperature for 30 min, and (c) Pd/C recovered from catalytic furfural hydrogenation of 30 mM furfural in 0.1 M phosphate buffer solution (pH 4.5) at 1 bar H₂ and room temperature for 1h.

Reviewer #2 (Remarks to the Author):

This report compares the validity of two reaction mechanisms for the hydrogenation of furfural to furfuryl alcohol. The first is a classical Langmuir-Hinshelwood (LH) mechanism based on hydrogen dissociation on the surface of Pd nanoparticles, and stepwise addition of adsorbed hydrogen to adsorbed furfural. The second is a proton-coupled electron transfer (PCET) mechanism.

The second is discarded early on because increases in hydronium concentration by 100,000 times, only increase the reaction rate by ~10 times. This is inconsistent with the kinetics of PCET.

In the second part of the report, the LH mechanism is evaluated in detail to identify the most important surface species, the rate-determining step, and the effect of pH on the hydrogen binding energy over Pd. Understanding this effect and the heat of adsorption of furfural on Pd, are key to explaining why reaction rate increases by lowering pH.

I think the conclusions of the report are solid and based on new information and kinetic analysis that are original to this report.

I am not strongly enthusiastic about recommending this report for Nature Communications because the approach to solving this mechanistic challenge is very classical; very well done and detailed, but not new in itself. The reaction itself is not particularly important; furfuryl alcohol has been produced without problems from furfural for many decades. And in heterogeneous catalysis extrapolation to other organic species is an educated guess, at best.

On the other hand, electrochemical processes and heterogeneous catalysis often occur simultaneously, and the individual contributions to observed reactivity can be disentangled by following variations of the research plan implemented in this report. In this sense, this report will be useful to the heterogeneous catalysis community as a blueprint to answer similar or related questions about catalysis and the metal-aqueous interface in the future.

The report could be published as is, but I found both the abstract and the introduction hard to follow. They deserve to be edited for clarity. In the first, the sentences do not follow each other logically and the report's key findings are lost. In the second, the paragraphs do not follow each other very logically (consider starting with the last one). I found the rest of the report clear and concise.

Response:

We appreciate the reviewer's supportive and valuable comments. As suggested, we revise the abstract and introduction for better logical flow (as highlighted in the revised manuscript). In addition, we emphasize the scientific significance of this study and new insights into electrochemical process. This study presents a comprehensive analysis of the bimolecular surface reaction, where the solvent effects on reaction rate and product selectivity are of broad interest to researchers in chemistry. We hope that this revised manuscript with clarification on the novelty and significant readership, improved logical flow, and enriched discussion that is grounded in new evidence suffice for publication in Nature Communications.

Reviewer #3 (Remarks to the Author):

The article » Impact of hydronium ions on the Pd-catalyzed furfural hydrogenation « by Yu et al. presents a mechanism study of furfural aldehyde hydrogenation in aqueous media. Sequential addition of hydrogen atoms by Langmuir-Hinshelwood mechanism is proposed and confirmed experimentally with variation of pH and isotope labeling, as well as in silico. This paper contradicts the proton-coupled electron transfer (PCET) mechanism recently published in Nature Catalysis. The methodology is rather standard, while it serves its purpose well. Topic is interesting for the catalytic community, however its relevance for the Nature Communication audience too specific and the impact of the research is rather limited, maybe a publication in Nature Catalysis will fit more, however this is a ultimate authority of the editor to decide. Serious flaw of the current article are suspiciously low activation barriers reported for diffusion-controlled reactions in water solution. Should the editor find this paper sufficiently relevant for wide audience of Nature Communication, I would consider this article suitable for publishing after a major revision. Careful addressing of comment 5 on the activation barriers will be crucial for the eventual acceptance.

Response:

We sincerely thank the reviewer for the valuable suggestions, which help us to improve the manuscript and to design future research studies. Please find our point-to-point responses below.

1. On page 12, it is claimed that only 50% of remaining sites are available for hydrogen adsorption due to the fact that furfural is only blocking these sites and are independent of furfural adsorption and do not compete for the same sites. This seems to be an overzealous assumption to assume that they are non-competitive as there are only metallic Pd sites available on the Pd/C catalyst. The presented model also only assumes that 1 molecule of furfural binds to 1 site which is not necessarily true.

Response:

We did not assume the binding of furfural to one Pd. In the presence of saturated furfural, we observed 50% decrease in the H₂ adsorption rate (Figure 7). Therefore, we infer that furfural at saturation occupy 50% of the total available sites that are originally accessible to H₂. As such, under hydrogenation condition (30 mM furfural, saturated adsorption), H₂ can access to only half of the sites. While this remains a semi-quantitative evaluation, we did not assume the quantity of Pd atoms per site. In addition, in our adsorption model (Figure 6), we actually consider furfural coverages on more than one Pd atom with reference to the previous study:

- Wang, S., Vorotnikov, V. & Vlachos, D. G. Coverage-induced conformational effects on activity and selectivity: hydrogenation and decarbonylation of furfural on Pd (111). *ACS Catal.* **5**, 104-112 (2015).

2. Why were the Pd catalysts pretreated in H₂ at room temperature? How can you assure that all of the Pd metallic species were reduced? This may also affect the total number of sites that were considered available for furfural and/or H₂ adsorption. With no TPR or XPS data presented for the Pd/C catalyst, it is possible that there are different amounts of reduced and oxide sites available at each of your reaction temperatures between 25 – 100 °C after reduction in situ.

Response:

The Pd/C in-situ pretreatment was performed at a high H₂ pressure of 30 bar. The XPS analysis indicates that Pd⁰ dominates the catalyst surface (Pd⁰:Pd²⁺ ratio increases from 0.88:1 to 4.4:1 after in-situ pretreatment; Figure S5a&b, see the copy below). The small amount of Pd²⁺ results from oxidation by exposure to air during catalyst recovery and storage prior to XPS analysis. Previous TPR analyses showed that the reduction of PdO is readily feasible at room temperature:

- Ferrer, V., Moronta, A., Sánchez, J., Solano, R., Bernal, S. and Finol, D., 2005. Effect of the reduction temperature on the catalytic activity of Pd-supported catalysts. *Catal. Today*, 107, 487-492.
- Chou, C.W., Chu, S.J., Chiang, H.J., Huang, C.Y., Lee, C.J., Sheen, S.R., Perng, T.P. and Yeh, C.T., 2001. Temperature-programmed reduction study on calcination of nano-palladium. *J. Phys. Chem. B*, 105(38), 9113-9117.

Figure S5. XPS curves and fittings of (a) as-received Pd/C, (b) Pd/C after in-situ aqueous-phase reduction at 30 bar H₂ and room temperature for 30 min, and (c) Pd/C recovered from catalytic furfural hydrogenation of 30 mM furfural in 0.1 M phosphate buffer solution (pH 4.5) at 1 bar H₂ and room temperature for 1h.

3. Does hydrogen partial pressure really affect the equilibrium coverage according to $n=0.5$? H₂-TPD (with pulse-chemisorption) experiment could demonstrate the validity of Eqn. S1a.

Response:

The adsorption equilibrium equation (Eqn. S1a) describes the reaction $[0.5 H_2 + * \rightleftharpoons H^*]$ where $n = 0.5$ based on the principle of mass balance.

$$K_{H_2}^{0.5} = \frac{\theta_H}{P_{H_2}^{0.5} \theta_{*,1}} \quad (\text{Eqn. S1a})$$

From this equation, hydrogen coverage is affected by H₂ pressure as long as saturation not reached. In our experiment, hydrogen at surface was far from saturation, thus, is affected by H₂ pressure.

4. It should be demonstrated and proven that reactor operates in kinetic regime. No effect of external and internal mass transfer limitations on the kinetic rate(s) should be determined either experimentally, via kinetic modeling or other criteria (e.g. Weisz–Prater).

Response:

Furfural mass transport is not rate-limiting, as a 0th reaction order of furfural was observed (Figure 4b).

H₂ diffusion is also not rate-limiting. We measured the adsorption rate of H₂ on Pd/C through the H/D exchange experiment, i.e., $H_2 (g) + D_2O (l) \rightarrow HD (g) + HDO (l)$, $0.5 H_2 (g) + D_2O (l) \rightarrow 0.5 D_2 (g) + HDO (l)$, (TOF > 12,000 h⁻¹; Figure 7). This indicates that the diffusion rate of H₂ and the collision frequency of H₂ on Pd is at least 12,000 h⁻¹, which is much higher than furfural hydrogenation (TOF < 2,500 h⁻¹; Figure 1b). Therefore, we conclude that H₂ diffusion does not limit the hydrogenation rate.

These justifications are included in the revised Supplementary Note S2.

5. Page 7: Apparent activation barriers of 9.9 and 13.8 kJ mol⁻¹ are very suspicious. In water solutions, diffusion-controlled reactions proceed with approx. 10⁹ s⁻¹, which is the highest reaction rate for a reaction in water solution. The corresponding diffusion barrier is 21.8 kJ mol⁻¹ from Eyring-Polanyi equation. This barrier is determined from the mobility of water molecules (water diffusion coefficient). The only exceptions are reactions of proton hopping between two adjacent water molecules or water molecule chains (Grotthuss mechanism of “proton wire”). Basically, all reactions with intrinsic barriers lower than 21.8 kJ mol⁻¹ will be diffusion controlled. If the authors claim that their barriers are lower, they need to justify this mechanistically. Even arguing that furfural is hydrogenated on the surface, which is true, does not circumvent this problem since furfural *must* be delivered to the surface through water, hence being subject to the diffusion barriers.

Response:

Thank you for the detailed explanation. We would like to clarify that the intrinsic energy barriers were determined to be far higher than 21.8 kJ mol⁻¹ (i.e., 41.4 kJ mol⁻¹ at pH 1.6 vs. 54.1 kJ mol⁻¹ at pH 5.8; Figure 11). Following your comment, furfural hydrogenation is concluded not to be diffusion-controlled. The conclusion is based on the experimental results (0th reaction order of furfural and high H₂ adsorption rate) as elaborated in the response to Comment 4 above.

6. Page 7: The barrier is not the energy level difference between the transition state and the ground state because these are two different things. Transition states are saddles on the potential energy surface, which the reaction starting at the initial state must overcome. The ground state means that there is no electron excitation, as opposed to excited state.

Response:

Thank you for the suggestion. The term “ground state” is changed to “state of adsorbed furfural” in this revision for more precise description (Page 7).

7. Page 8: In Ref [21], -23 kJ mol⁻¹ is the free energy of adsorption of FAL and 2 H* and not only FAL is the authors claim. This means that they cannot compare the value -19 kJ mol⁻¹ directly. Moreover, Gibbs free energy is *a strong* function of temperature, which the authors do not state. While the Gibbs free energy can be as low as -19 kJ mol⁻¹, the temperature-independent adsorption energy (or enthalpy) must be stronger (=more negative).

Response:

We agree with the reviewer in this point. As the comparison is no longer meaningful, we removed it from the statement and clarified the temperature:

“This corresponds to a standard Gibbs free energy (ΔG°) of adsorption of about -19 kJ mol⁻¹ at room temperature.” (Page 8)

The adsorption enthalpy of furfural on Pd is approximately -30 kJ mol⁻¹ at saturation (Figure 5b), in good agreement with the reviewer’s estimation.

8. Page 8: The authors state the furfural-Pd bonds are strong and remain unaffected by H₂O-involving interaction. The very same Ref [21] contradicts this on Page 18 of the supplementary by stating: »Furfural adsorption energy in the gas phase was taken from experimental TPD values of -0.98 ± 0.20 eV[16,17].« This is a change from 100 kJ mol⁻¹ to 19 kJ mol⁻¹, which is not *unaffected*.

Response:

We intended to compare furfural-Pd bond strengths at varying pH in aqueous phase and not to compare the bond strength in gas phase against that in aqueous phase. Obviously, the former is significantly stronger, also because adsorption occurs from the solvated state. To clarify, the sentence is revised as

“...the furfural-Pd bond in aqueous phase remains unaffected by electrochemical changes associated with the increasing hydronium ion activity”. (Page 8)

9. In fact, the authors admit this on Page 9: »Without the influence of water (i.e., in ultrahigh vacuum (UHV)), the furfural adsorption enthalpy ($\Delta H_{\text{fads}}^{\text{g}}$) was computed to be about -164 kJ mol⁻¹ (-1.7 eV) if furfural lies flat on a clean Pd surface at a low density of 0.25 monolayer (ML) [14].« and »The enthalpy changes to 179 about -97 kJ mol⁻¹ as the surface becomes crowded at 0.5-1 ML and furfural binds to Pd via the – CHO group, resulting in a tilted conformation.[14]«

Response:

We revised the statement as in the response to Comment 8 above.

10. Page 9: I am not sure that talking about one side of the molecule being solvated makes sense. Solvation (or adsorption on Pd) is a complex process that entails significant electron density redistribution.

Response:

This approach has been inspired by the work from Singh and Campbell (2019), who used the same solvation model and successfully explained the adsorption of phenol on Pt(111) in water. Considering that both phenol and furfural are aromatic compounds, we adopted the same model for furfural solvation in the current study in order to allow a reader to note the similarity of the processes.

- Singh, N. & Campbell, C. T. A simple bond-additivity model explains large decreases in heats of adsorption in solvents versus gas phase: a case study with phenol on Pt (111) in water. *ACS Catal.* **9**, 8116-8127 (2019).

We agree that solvation is significantly more complex and may follow more complex patterns and are in the process of addressing this complexity at present with theoretical approaches.

11. Page 9: Why is one fourth of furfural area blocked by adjacent furfural molecules? Provide justification.

Response:

One fourth of surface area blockage has been an assumption to close the thermodynamic cycle. After revisiting our adsorption model, we removed this assumption from the revised manuscript. The discussion and Figure 6 were revised accordingly.

“The change from state (A) to (E_{flat}) or (E_{tilted}) is the adsorption of gas furfural on Pd in the flat or tilted mode, respectively. The DFT calculated enthalpy ($\Delta H_{\text{f ads,g}}$) of parallel and tilted adsorption was about -164 kJ mol^{-1} and -97 kJ mol^{-1} , respectively.²⁰ To close the thermodynamic cycle, changing from state (E_{flat}) to (D) would be associated with an enthalpy $\Delta H_{\text{f ads,solv,flat}}$ of $+36 \text{ kJ mol}^{-1}$. In comparison, changing from state (E_{tilted}) to (D) would decrease the enthalpy with $\Delta H_{\text{f ads,solv,tilted}}$ of -31 kJ mol^{-1} . The step from either (E_{flat}) or (E_{tilted}) to (D) is essentially the solvation of adsorbed furfural by liquid water, thus more reasonably to be exothermic. Therefore, adsorbed furfural molecule is more likely to be in tilted geometry in the adsorption and reactions at saturated region.” (Page 9)

Figure 6 | Thermodynamic cycle for the gas- and aqueous-phase adsorption of furfural (F) on Pd. A replacement ratio of 6.5 water molecules by one furfural molecule on Pd is used. Ultrahigh vacuum above the clean Pd surface is denoted as UHV/Pd. For simplicity, the difference between bond energy (ΔU) and enthalpy (ΔH) in gas-forming steps is considered negligible ($\Delta U = \Delta H + RT$, and $RT = 2.5 \text{ kJ mol}^{-1}$ at 298 K is omitted). ^a Solvation enthalpy calculated using the van't Hoff equation and Henry's law constant^{27,28}. ^b Enthalpy of adsorption of water on Pd (see **Supplementary Note S1**). ^c Enthalpy of saturated adsorption in aqueous phase measured by liquid colorimetry. ^d Computed adsorption enthalpy at 0.25 ML and ^e at 0.5-1 ML in UHV from ref ²⁰.

Reviewer #4 (Remarks to the Author):

The mechanism of the selective hydrogenation of polar carbonyl groups in aqueous phase and the effect of pH on the reaction rate are intensely debated. Several groups have proposed water-mediated hydrogenation steps to be involved. Using detailed kinetic and isotope measurements, the authors claim to show “unequivocally” that the hydrogenation occurs via sequential addition of hydrogen atoms via a Langmuir Hinshelwood mechanism. These are carefully executed experiments, but the evidence is unfortunately not sufficient to convince me.

Response:

We appreciate the reviewer’s time on the careful examination of our manuscript. We performed supplementary experiments (e.g. low-concentration furfural adsorption and catalytic hydrogenation, XPS, XRD), presenting additional evidence (e.g. Figure 1a, 5b, S2, S4-6; Table S1, S2) to substantiate our findings.

The key experiment is shown in Figures 2 and 3. When H₂ is replaced D₂, the TOF decreases significantly and FAL product mass spectra also shift up. While the H-atom that hydrogenates the C atom in RCHO is generally accepted to originate from adsorbed H, the debate is about the H atom that attacks the O atom. Unfortunately, FAL alcohol rapidly exchanges its proton with water (Figure 3a) and the origin of that H cannot be determined with isotope experiments. The increase in m/z 100 from D₂/H₂O to D₂/D₂O actually suggests that this “H” comes from water.

Response:

We agree with the reviewer that the origin of H that attacks the carbonyl O atom cannot be determined. The writing was revised to clarify that Langmuir Hinshelwood mechanism applies to the H addition to the carbonyl C atom, e.g., in the Abstract:

“Instead of a proton-coupled electron transfer pathway, our results show that a Langmuir-Hinshelwood mechanism describes the rate-limiting hydrogen addition step, where hydrogen atom adsorbed on Pd is transferred to the carbonyl C atom of the reactant.”

Note that H attacking C tends to be the *second* H addition step according to Zhao et al., and it is the rds according to the observed 1st reaction order of H₂ (Figure 4a). That means, the *first* H addition step, i.e., H attacking O, is in the state of quasi-equilibrium rather than rate-limiting. Thus, it cannot be determined whether or not this first step occurs via a proton coupled electron transfer. In consequence, the mechanism of the first hydrogen addition does not impact the rates (and TOFs) observed in the current study.

- Zhao, Z. *et al.* Solvent-mediated charge separation drives alternative hydrogenation path of furanics in liquid water. *Nat. Catal.* **2**, 431-436 (2019).

Hydrogen adsorption is not kinetically relevant, as the authors indicate and is suggested by the first order kinetics in H₂. Figure 10 shows that the kinetically relevant energy difference is between F* + H₂(g) and the (FAL) transition state. The energy diagram in Figure 10 is somewhat unexpected since weaker H* adsorption (and hence a less favorable hydrogenation reaction step since the product FAL(l) does not change) corresponds with a more stable transition state and a lower activation energy. A BEP relation for a typical LHHW elementary step would suggest the opposite trend, ie a higher activation energy for a less favorable reaction step.

Response:

We fully agree with the reviewer that H₂ adsorption is not a kinetic function but a state function in this study. The scenario described by the reviewer (higher activation energy, i.e., (FAL)[‡] energy level of pH 1.6 lying above that of pH 5.8; Figure 10) is significant for early transition state, however less sensitive for late transition state. In the energy profile (Figure 10, a copy below), enthalpy of transition state is much less affected by the stabilization/destabilization of adsorbed states, showing a late transition state.

Moreover, samples and strategies to break the BEP relationship (the linear scaling relationship) have been explored. Our own recent works also give two samples in alcohol dehydration and in benzaldehyde hydrogenation, respectively:

- Cheng, G. *et al.* Critical role of solvent-modulated hydrogen-binding strength in the catalytic hydrogenation of benzaldehyde on palladium. *Nat. Catal.* **4**, 976-985 (2021).
- Pfriem, N. *et al.* Role of the ionic environment in enhancing the activity of reacting molecules in zeolite pores. *Science* **372**, 952-957 (2021).

Figure 10 | Energy diagram for the illustration of pH effect on hydrogenation kinetics.

The argument for tilted F adsorption via the RCHO group are based on the thermodynamic cycles in Figure 6. These calculations are based on experimental data points and several assumptions and not fully conclusive. Spectroscopic data could provide more direct proof for the geometry. According to the thermodynamic cycle in Figure 6, parallel adsorption should be more favorable, but might not occur at high coverage. From these arguments, one would expect a change in selectivity and in heat of adsorption for very low F concentrations, but this is not observed.

Response:

We agree with the reviewer's hypotheses and performed additional adsorption experiments. The newly added Figure 5b shows higher furfural adsorption heat at low furfural concentrations (new discussion in Page 10).

In addition, the hydrogenation selectivity shifts from the carbonyl hydrogenation product (FAL) to the ring hydrogenation product (THFF) when the furfural concentration decreases from 30 mM to 1.3 mM as shown in Figure S2 and Table S2. The ratio of ring hydrogenation (THFF) to C=O hydrogenation (FAL) is about 0.15/1 at furfural concentration of 30 mM, while the ratio increased to about 0.8/1 at concentration of 1.3 mM.

Figure 5b | Adsorption enthalpy for furfural adsorption on Pd/C as a function of furfural concentration in aqueous phase. The measurement was performed using liquid calorimetry at room temperature.

Table S2. Selectivity of furfuryl alcohol (FAL), tetrahydrofurfural (THFF), and tetrahydrofurfuryl alcohol (THFAL) from the hydrogenation of furfural at low and high concentration.*

pH	Initial furfural concentration (mM)	 Furfural conversion (mol%)	Product selectivity (mol%)		
			 FAL	 THFF	 THFAL
4.5	1.3	66.5	30.7	23.5	45.9
		96.0	26.2	21.1	47.3
4.5	30	34.0	74.2	10.7	5.6
5.8	30	50.9	83.1	12.1	7.8

* The reaction was performed in 0.1 M phosphate buffer solution at 1 bar H₂ and room temperature.

Figure S2. Furfural and product concentrations as a function of reaction time during hydrogenation at a low initial furfural concentration of 1.3 mM. The reaction was performed in 0.1 M phosphate buffer solution (pH 4.5) with 1.4 mg Pd/C at 1 bar H₂ and room temperature.

Kinetically, it is very difficult to distinguish competitive and non-competitive adsorption. Since F adsorbs stronger than H₂, it is expected that the F coverage is saturated and H₂ adsorbs at the available sites between F molecules. At very low F concentrations, one would however expect to see kinetic effects of competition between F and H₂.

Response:

We agree with reviewer that kinetic effects of competitive adsorption are expected at low F coverage. We also note here that a series of changes, besides this competition, occur at low F concentration that complicated the reaction. The supplementary experiments present evidence of changes in hydrogenation product selectivity associated with furfural concentration (Figure S2, Table S2). These are preliminary data implying the possible change in adsorption coverage and orientation as the bimolecular system shifts from competitive adsorption to *non*-competitive adsorption. Moreover, the hydrogenation of furan ring consumes four H atoms while that of carbonyl takes only two H atoms. This is expected to impact furfural conversion and H₂ adsorption in a complex way. Therefore, we will conduct a standalone study in the future for more comprehensive investigation, to untangle the correlations among adsorption coverage, binding mode, competitive/non-competitive adsorption, product distribution, and reaction kinetics.

REVIEWERS' COMMENTS

Reviewer #1 (Remarks to the Author):

The revised version of the manuscript has clarified, to a very good extent, all the points I have raised in my first review. The new information (both in terms of explanations but also in terms of new data) improved substantially the manuscript which I now believe is suitable for publication in Nature Communications.

Reviewer #2 (Remarks to the Author):

The authors have responded to the reviewers' comments in detail and have addressed their concerns in the newly revised manuscript. I recommend the publication of the revised manuscript as is.

Reviewer #3 (Remarks to the Author):

The authors substantially improved the manuscript, which is now suitable for publication. Regarding the answer to my (reviewer 3) question 6, I have the following remark: Add "initial state" in parentheses. The description should be "co-adsorbed furfural and H* (initial state)".

Reviewer #4 (Remarks to the Author):

The authors have addressed my comments in detail and performed several additional experiments. I now fully agree with the abstract, and the response to my first comment, that the rds is the addition of a surface hydrogen atom to the carbonyl C atom. I also agree with the reply (which is unfortunately not included in the manuscript) that nothing can be said about the mechanism of H attacking O, as it is a quasi-equilibrated step. It remains highly likely that this is a PCET step, not a LH step, as the authors still suggest in step 3 of their kinetic model.

It further seems that the non-BEP relation in Figure 10 could in part be attributed to the change in the stability of the adsorbed FH* intermediate, which is also affected by the pH. The elementary step is the hydrogenation of FH*, which could still follow a BEP relation. With this refinement, the proposed model is now in line with recent DFT studies of the carbonyl hydrogenation mechanism.

Reviewer #1 (Remarks to the Author):

The revised version of the manuscript has clarified, to a very good extent, all the points I have raised in my first review. The new information (both in terms of explanations but also in terms of new data) improved substantially the manuscript which I now believe is suitable for publication in Nature Communications.

Response: We appreciate the Reviewer's valuable comments.

Reviewer #2 (Remarks to the Author):

The authors have responded to the reviewers' comments in detail and have addressed their concerns in the newly revised manuscript. I recommend the publication of the revised manuscript as is.

Response: We appreciate the Reviewer's valuable comments.

Reviewer #3 (Remarks to the Author):

The authors substantially improved the manuscript, which is now suitable for publication. Regarding the answer to my (reviewer 3) question 6, I have the following remark: Add "initial state" in parentheses. The description should be "co-adsorbed furfural and H* (initial state)".

Response: We appreciate the Reviewer's valuable comments. The term "(initial state)" was added as suggested (Page 6).

Reviewer #4 (Remarks to the Author):

The authors have addressed my comments in detail and performed several additional experiments. I now fully agree with the abstract, and the response to my first comment, that the rds is the addition of a surface hydrogen atom to the carbonyl C atom. I also agree with the reply (which is unfortunately not included in the manuscript) that nothing can be said about the mechanism of H attacking O, as it is a quasi-equilibrated step. It remains highly likely that this is a PCET step, not a LH step, as the authors still suggest in step 3 of their kinetic model. It further seems that the non-BEP relation in Figure 10 could in part be attributed to the change in the stability of the adsorbed FH* intermediate, which is also affected by the pH. The elementary step is the hydrogenation of FH*, which could still follow a BEP relation. With this refinement, the proposed model is now in line with recent DFT studies of the carbonyl hydrogenation mechanism.

Response: Thank you for your suggestions. As the pathway of 1st H addition cannot be confirmed in this study, we revised Step 3 in Table 2 (a copy below), which now includes the LH step, PCET, and Volmer step. Nevertheless, the our kinetic study remains valid regardless of the pathway of 1st H addition because it is quasi-equilibrated. The manuscript is revised accordingly:

“...Table 2 compiles the elementary steps: (1) dissociative adsorption of H₂ on the active site to two H*; (2) adsorption of a furfural molecule; (3) the first H addition to the adsorbed furfural molecule (both the LH pathway and/or via PCET and the reverse Volmer step); and (4) the second H addition to the adsorbed furfural-H intermediate (via the LH pathway). The reaction order with respect to furfural concentration is zero, thus, Step 2 is quasi-equilibrated (Figure S1b), whereas the 1st order in H₂ suggests the quasi-equilibrium established in Step 3 (Supplementary Note S4). Although our KIE study presents no evidence of the H addition

pathway in Step 3 (Figure 2), it remains sensible to account the surface species quasi-equilibrated with constant K_1^o in our kinetic model because all species in this step are in the state of quasi-equilibrium regardless of pathway. Finally, Step 4 is the rds and it involves the addition of a surface H^* atom to the carbonyl C of furfural, according to the KIE results (Figure 2). ...”

Table 2. Elementary steps in furfural hydrogenation.

(Step 1) H_2 adsorption	$H_2 + 2 * \rightleftharpoons 2 H^*$	$K_{H_2}^o$
(Step 2) Furfural adsorption	$F + * \rightleftharpoons F^*$	K_F^o
	$F^* + H^* \rightleftharpoons FH^* + *$	K_1^o
(Step 3) 1 st H addition	$F^* + H^* + e^- \rightleftharpoons FH^*$	$K_{1,PCET}^o$
	$H^* \rightleftharpoons H^+ + e^- + *$	$K_{1,Volmer}^o$
(Step 4) 2 nd H addition	$FH^* + H^* \rightarrow FH_2 + 2 *$	k_2

We agree that FH^* could be affected by pH, although this cannot be quantitatively examined in this study. The simplified yet defensible energy diagram (Figure 7) is therefore presented for the illustration of pH effect on H^* binding and its kinetic consequence.